# Filter-Feeding Pacific Lamprey (*Entosphenus tridentatus*) Ammocetes Can Reduce Suspended Concentrations of *E. coli* Bacteria

Parker Kalan [1,2,*], John Steinbeck [2], Freddy Otte [3], Sean C. Lema [1] and Crow White [1]

1 Biological Sciences Department, Center for Coastal Marine Sciences, California Polytechnic State University, San Luis Obispo, CA 93407, USA
2 Tenera Environmental, Inc., San Luis Obispo, CA 93401, USA
3 Natural Resources, San Luis Obispo, CA 93401, USA
* Correspondence: parkerkalan@gmail.com

**Abstract:** Filter-feeding invertebrates such as bivalves have been shown to improve the health of aquatic systems by reducing concentrations of bacteria and other harmful suspended organisms, but it remains unknown if microphagous suspension-feeding fishes can provide similar ecosystem services for water quality. Here, we tested whether the presence of the filter-feeding larval ammocoete life-stage of Pacific lamprey (*Entosphenus tridentatus*) can reduce suspended concentrations of *Escherichia coli* bacteria. Aquaria containing either filter-feeding ammocoete lamprey larvae (1.5 fish·L$^{-1}$), lamprey macropthalmia juveniles (1.5 fish·L$^{-1}$) that do not suspension-feed, or no lamprey (control) were filled with water contaminated with *E. coli* bacteria and then monitored for 5 d for *E. coli* concentration changes in the water column. The presence of ammocoete larvae generated a significantly faster decline in *E. coli* abundance compared to aquaria containing either macropthalmia-stage lamprey or no fish, which showed similar *E. coli* concentration profiles over that 5 d period. A higher density of ammocoetes (4.3 fish·L$^{-1}$) resulted in a more rapid decline in *E. coli* compared to the lower 1.5 fish·L$^{-1}$ ammocoete density, further implying that ammocoetes augmented bacterial clearance. These observations provide evidence that filter-feeding larval ammocoetes of Pacific lamprey may help promote water-quality enhancement by reducing suspended bacterial concentrations.

**Keywords:** fish; Petromyzontiformes; suspension feeding; water quality; coliform; bacteria; pollution; bioremediation

## 1. Introduction

Maintaining high water quality in rivers and streams is among the primary goals in water-resource management, as healthy freshwater systems are critical both for human use of these systems for water consumption, recreation, and fishing and for supporting communities of native aquatic species [1–3]. It is well established that the presence of filter-feeding and detrital-processing aquatic invertebrates, such as bivalve molluscs, in aquatic systems can serve as a nutrient-reduction tool for bioremediation and, thus, can be an important component of achieving resource-management goals for freshwater and marine systems [4,5]. Filter-feeding bivalves, in particular, have been shown to help control the overabundance of phytoplankton resulting from eutrophication when freshwater systems experience excess inputs of nitrogen and other nutrients from terrestrial sources [6]. The maintenance or restoration of filter-feeding organisms in a freshwater system thus has potential to provide a crucial ecosystem service by improving water quality, even in aquatic systems highly impacted by human activity [7].

While bivalve molluscs such as oysters and clams have been the primary focus of studies examining the ecosystem services provided by filter-feeding organisms in watersheds [8,9], other filter-feeding species also have the potential to support water-quality

management, especially in freshwater systems without high densities of filter-feeding invertebrates. One such example is the Pacific lamprey (*Entosphenus tridentatus*, order Petromyzoniformes), an anadromous agnathan fish that historically inhabited much of the Northern Pacific Ocean from Japan to Baja California, Mexico [10]. Pacific lamprey evolved a parasitic adult stage that lives in the open ocean for multiple years and then migrates up coastal rivers and streams to spawn in a benthic nest, or redd [11,12]. After eggs from that redd hatch, the larval life-stage—called an ammocoete—resides in the home stream or river with its body buried in fine sediments (i.e., silt beds, mud) but mouth positioned at the sediment–water interface to allow for microphagous suspension-feeding from the water. Ammocoetes may remain filter-feeding in the freshwater environment for up to seven years before metamorphosing into a non-filter-feeding juvenile (macropthalmia) stage that migrates downstream to the ocean. During this extended developmental period in the freshwater environment, specialized pharyngeal morphology of ammocoetes allows for low-velocity pumping of water over the gills [13], where suspended microorganisms, detritus, and algae are trapped in a mucosal membrane and then transported to the digestive tract [13–16]. While the rate of water movement over the feeding structures of lamprey ammocoetes was documented as the lowest recorded for any suspension-feeding organism, suggesting that ammocoetes required high concentrations of suspended organic particulates [13], the similarity of their suspension-feeding to that of filter-feeding invertebrates suggests that Pacific lamprey may play an ecological role in removing bacteria from the freshwater environment, either via direct capture in the pharyngal mucus or via collection of organic particulates with attached bacteria.

Importantly, lamprey ammocoetes may provide an ecosystem service in freshwater systems by reducing concentrations of harmful bacteria, including *Escherichia coli* and associated fecal pathogenic bacteria [17]. Concentrations of suspended coliforms and bacteria in the digestive tracts and fecal waste of animals, are commonly quantified by water resource managers as a bio-indicator for the presence of pathogenic strains of bacterial or fecal pollution. Fecal coliforms, in particular, are a leading cause of impairment to surface-water quality [3,18–20] and can present a human health hazard if consumed or if exposure occurs at high concentrations [21]. This bacterium enters streams from agricultural fields, sewage overflows, direct defecation by animals and people, and surface runoff [22], and increased levels of *E. coli* and associated fecal coliforms in a watershed have been shown to have adverse effects on biodiversity and ecological health [23].

In light of the established role of filter-feeding invertebrates in reducing suspended bacterial concentrations [7], it is possible that filter-feeding ammocoetes may contribute to ecosystem health by reducing concentrations of bacteria in watersheds. However, the effects of ammocoetes on pathogenic bacterial load—and, specifically, coliform *E. coli* concentrations in water—have not been tested. Here, we examined the potential for filter-feeding ammocoetes to reduce bacterial loads in a freshwater environment by exploring how the presence of ammocoetes affected suspended *E. coli* load in a closed-system, controlled environment. Specifically, Pacific lamprey ammocoetes collected from the wild were placed into experimental aquaria containing water sourced from a creek system containing *E. coli*, and the concentration profile of suspended *E. coli* was then monitored over 5 d and compared to aquaria containing either non-filter-feeding lamprey macropthalmia at the same fish density or no fish at all (control) to test the hypothesis that the presence of suspension-feeding ammocoetes can reduce pathogenic bacterial concentrations. The purpose of this research was to provide additional understanding about the filter-feeding life-history phase of Pacific lamprey and to promote the species ecological role in watersheds, especially through a management/water-improvement lens.

## 2. Methods

### 2.1. Lamprey Ammocoete Collection

Pacific lamprey (*Entosphenus tridentatus*) ammocoetes (*n* = 178) were collected on 30 July 2019 from the Carmel River near Carmel, CA, USA, approximately 9 km upstream

from the river outlet to the ocean (36°31′34.0″ N 121°50′27.2″ W) (California Department of Fish and Wildlife, Scientific Collecting Permit 10287). Collection was done with an APB-2 (Engineering Technical Services, Madison, WI, USA) electrofishing backpack to coax ammocoetes out of sediment and into the water column (low-burst pulse, rate 3.00 pulses/s, 25% duty cycle, voltage 125–200 V), where they were quickly hand-netted. Collected ammocoetes (body length range: 2–16 cm; body mass range: 0.20–4.20 g) were placed in coolers containing aerated river water maintained at ~15 °C using ice packs, and transported immediately by vehicle to the San Luis Obispo city Wastewater Treatment Facility in San Luis Obispo, CA, USA. During the initial acclimation and subsequent experimental periods of captive maintenance, ammocoetes exhibited active behaviors, such as burrowing and moving their mouths, indicative of them being in good health.

### 2.2. Fish Acclimatization Period

Ammocoetes were placed in two 475 L holding aquaria (183 cm × 46 cm × 71 cm) at the San Luis Obispo Water Reclamation Facility. These holding aquaria were pre-filled with ~350 L unfiltered water collected from the urban region (35°14′40.1″ N 120°40′47.2″ W) of San Luis Obispo Creek in San Luis Obispo, CA, USA. The bottom of each aquarium contained 12 cm of soft sediment collected from the same location as water collection in San Luis Obispo Creek. Ammocoetes were maintained in these aquaria for 12 weeks prior to the start of the experiment to allow for acclimatization to laboratory conditions. During this acclimatization period, water was recirculated through the holding aquaria using a standard aquarium filter. During this time fish were not fed; however, because the holding tanks contained unfiltered creek water and sediment that were a viable environment with abundant nutrients to sustain the ammocoetes during their acclimatization period. Between the time of collection and commencement of experimentation, 43 of the 178 (24%) ammocoetes metamorphosed into macropthalmia, developing eyes and teeth; those fish were used for the macropthalmia treatment condition in the experimental design described below.

### 2.3. Experimental Treatments

Three experimental treatments were evaluated, each represented by the number of individual lampreys of a particular development stage: ammocoete low density ('Ammocoete low' treatment: $n$ = 20 lamprey/aquarium for ~243 fish·m$^{-2}$ substrate; total fish wet biomass: 19.26 ± 2.57 g (mean ± SD)), macropthalmia ('Macropthalmia' treatment: $n$ = 20 lamprey per aquarium for ~243 fish·m$^{-2}$; total biomass: 29.07 ± 11.20 g), and a 'Control' condition with aquaria containing no lamprey ($n$ = 0). The 'Ammocoete low' treatment was replicated across four aquaria, the 'Macropthalmia' treatment was replicated across two aquaria, and the 'Control' was replicated across five aquaria. In addition, a high density of ammocoetes ('Ammocoete high' treatment: $n$ = 57 lamprey per aquarium for ~692 fish·m$^{-2}$; total biomass: 40.05 g) was also evaluated to provide a picture of how a higher ammocoete density might affect bacterial concentration conditions; however, due to the limited number of ammocoetes available, this 'Ammocoete high' treatment was not replicated and was conducted only in a single aquarium.

On 9 December 2019, Pacific lamprey individuals were transferred from the 475 L acclimation aquaria to a 22 L sorting aquarium and transferred into corresponding treatment aquaria. During this process lamprey individuals were placed on a tray, weighed, and then transferred to their respective 22 L experimental aquaria (20.3 cm × 40.6 cm × 26.7 cm). Prior to using the aquaria, each tank was sterilized, rinsed, and filled with 1.7 L of washed play sand. Then, 13.2 L of freshly collected creek water (see description below) was added to each aquarium, resulting in the following volume-based lamprey densities per tank in each treatment group: 'Ammocoete low', 1.5 fish·L$^{-1}$; 'Macropthalmia', 1.5 fish·L$^{-1}$; 'Control', 0 fish·L$^{-1}$; and the unreplicated 'Ammocoete high' tank with 4.3 fish·L$^{-1}$. Transfer of lamprey was conducted using hand nets to avoid injuring the fish. The laboratory space was below ground and windowless, so natural light was negligible; fluorescent ceiling lights were on a timer to generate a 16L:8D photoperiod. Experimental aquaria

were randomly assigned to positions on a single aquarium rack in the laboratory, and temperature in the tanks averaged ~15.5 °C during the experimental period.

## 2.4. Source Water for Experimental Treatments

Water used for the experimental treatments was collected from San Luis Obispo Creek in southern San Luis Obispo County along the central coast of California, USA. San Luis Obispo Creek originates in the Santa Lucia Mountains and flows approximately 13 km before entering the Pacific Ocean at Avila Beach, CA, USA. The San Luis Obispo Creek watershed supports an endangered population of anadromous rainbow trout [24] (i.e., steelhead, *Oncorhynchus mykiss*), and is currently thought to be the southern range limit for Pacific lamprey [25].

San Luis Obispo Creek is subject to urban run-off and a section of the stream flows through a diversion tunnel that is home to thousands of pigeons (Family Columbidae). Additionally, defecation by humans encamped in the city along the creek likely contributes contaminants to the creek, including possibly fecal coliforms [26]. These factors collectively contribute to degraded water quality and increased levels of fecal pathogenic bacteria in San Luis Obispo Creek, with the part of the creek that flow through an urbanized area showing substantially higher levels of *E. coli* and increased levels of total fecal coliforms [27]. Recent measurements of the total maximum daily load (TMDL) of pollutants in the urban region of San Luis Obispo Creek exceeded 9000 most probable number (MPN/100 mL), compared with a target level safe for human contact of 200 MPN/100 mL [3].

Water used in the experiment was collected from San Luis Obispo Creek with sterilized, rinsed carboys on 8 December 2019, shortly after a rainstorm. The collection site was in Mission Plaza (35°16′46.4″ N 120°39′52.1″ W) in the downtown city of San Luis Obispo, California, which is within the city's urban corridor. Water was transported immediately to the laboratory and added to the experimental aquaria, which were then stocked with lamprey as described above. *E. coli* in a water sample naturally lose viability if not provided with additional nutrient inputs, declining 100-fold in concentration of viable bacteria after approximately 4 to 11 d [28]. In this study, each tank was a closed system (no added nutrients), thus our a priori expectation was that the abundance of viable *E. coli* would decline to near zero in all our treatments in several days. We therefore focused on comparing the rate of decline of *E. coli* among the lamprey treatment conditions.

## 2.5. E. coli Quantification

We measured the relative viable concentration of suspended *E. coli* in the water column for each experimental aquarium once per day between 08:00 and 09:00 h from the start of the experiment until the concentration declined to less than 5% of the initially detected level in all experimental aquaria. *E. coli* concentration was quantified using the IDEXX Quanti-Tray System (IDEXX Laboratories, Inc., Westbrook, ME, USA). The IDEXX method is a standardized method used by the California Water Resources Control Board and other municipalities for quantifying viable *E. coli* concentration in a water sample [3].

IDEXX involves treating a 100 mL sample of water with ColiLert Reagent and transferring the water into a Quanti-Tray that has 24 small wells and 49 large wells. The tray is sealed and incubated at 30 °C for 24 h, allowing the reagent to bind to *E. coli* and fluoresce. The proportion of large and small wells that fluoresce provides an estimate of the most probable number (MPN) of colony-forming *E. coli*. If all wells fluoresce, a dilution factor is applied, and the MPN is calculated taking this dilution into consideration. For this experiment, *E. coli* concentrations were determined by collecting 10 mL of water from the middle of the water column in each treatment aquarium using a sterile pipette. That 10 mL water sample was then combined with ColiLert™ reagent and 90 mL sterile deionized water, creating a 1:10 dilution for analysis in the Quanti-Tray, because of the high concentration of bacteria.

*2.6. Statistical Analyses*

A Brown–Forsythe test was used to determine if the data met the assumption of homogeneous variances among the treatment levels. No significant differences in variance among the treatments was detected using the Brown–Forsythe test, so untransformed data were used in all analyses.

A one-factor repeated measures ANOVA was used to test for a significant difference in the rate of decline in *E. coli* concentration (MPN/100 mL) among the ammocoete (low density), macropthalmia, and control treatments over the experimental period. The first factor in the statistical analysis was lamprey life-stage (ammocoete, macropthalmia, or not applicable for the control treatment), which was treated as a fixed factor in the model. The second factor was time (number of days since the experiment began), which was the random factor in the model. Since MPN was quantified several times for each treatment (each day over five days), the data were analyzed as a repeated measures model, which accounts for potential correlations between consecutive measurements in a treatment over the study period. Data were analyzed using the Proc Mixed procedure in SAS (SAS Institute 2016), which supports analysis of mixed models that include both fixed and random factors. It also supports the evaluation of different variance structures to account for the repeated measures factor in the model, including autoregressive error structures that are common in studies using repeated measures [29]. The Proc Mixed procedure provides two model fitting criteria, the Akaike information criterion (AIC) and the Bayesian information criterion (BIC), to help in the selection of the appropriate error structure for the data [30]. Autoregressive, compound-symmetry, unstructured, and variance-component models of variance structure were evaluated in the Proc Mixed procedure analysis, and the outputs of each were compared based on their AIC and BIC values.

Pairwise tests between each treatment combination for each day of the experiment were conducted using the PDIFF option in the SAS Proc Mixed procedure, which compares mean values between treatment types between repeated measures. To account for the multiple number of tests, the degrees of freedom in the analyses used a Bonferroni correction. Data from the ammocoete high-density treatment are provided in this study but were not included in statistical analysis as the treatment condition was not replicated.

## 3. Results

The initial concentration of viable *E. coli* in the SLO Creek water in all aquaria at the start of the experiment averaged 241,960 MPN/100 mL, with no difference in initial *E. coli* concentrations among the treatment groups. After 5 d of the experimental treatment, *E. coli* concentration had declined in all replicate experimental aquaria to less than 5% of the initial MPN value, which was the pre-determined threshold for ending the experiment.

As expected, the concentration of viable *E. coli* declined in all of the aquaria (including 'control' condition aquaria) over that 5 d experimental period, but the average rate of decline in *E. coli* was significantly more rapid in the aquaria with ammocoetes than in 'control' tanks with no lamprey or in aquaria with the macropthalmia lamprey life-stage (Figure 1) (treatment × day interaction: $F_{10,8} = 14.03$, $p < 0.001$). Between days 1 and 2 of the experiment, the mean concentration of *E. coli* declined by 74.1% in the 'Ammocoete low' treatment aquaria, but by only 8.9% and 22.4% in the 'Macropthalmia' and 'Control' treatments, respectively. Those differing rates of *E. coli* decline led to statistically significant pairwise differences on day 2 between the ammocoete treatment and both the control and macropthalmia treatments (Table 1). No pairwise differences, however, were detected between the control and macropthalmia treatments on any of the sampling days. While not replicated, the concentration of *E. coli* declined most rapidly in the 'higher density' ammocoete tank, with an 89% reduction in *E. coli* concentration between sampling days 1 and 2 (Figure 1).

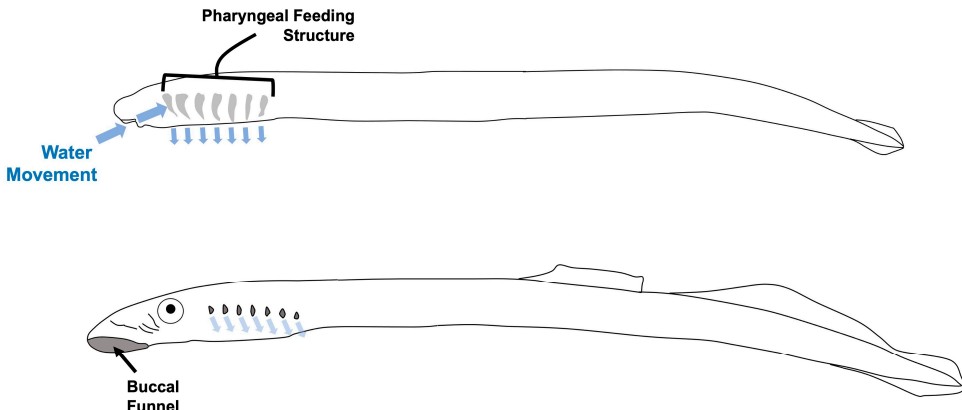

**Figure 1.** Lamprey ammocoete (**top**) and macropthalmia (**bottom**) morphology. Larval ammocoete move water over an internal pharyngeal pouch structure for microphagous filter-feeding. Eyed macropthalmia have a cup-like buccal funnel containing the mouth as well as several bicuspid teeth, an oral tooth plate, and tongue tooth, as well as seven pore-like gill openings on each side of the head.

**Table 1.** Pairwise comparisons of treatment effects on *E. coli* concentrations by sampling day. Values are estimated differences to illustrate the extent of variation within each pairwise comparison. Pairwise differences considered significant after Bonferroni correction ($p < 0.0033$) are highlighted in bold.

| Treatment Comparison | Day 1 | Day 2 | Day 3 | Day 4 | Day 5 |
|---|---|---|---|---|---|
| Control vs. Ammocoete | 13,734 ($p = 0.3734$) | **114,735 ($p = 0.0002$)** | 9213 ($p = 0.7362$) | 7907 ($p = 0.1460$) | 3784 ($p = 0.1233$) |
| Macropthalmia vs. Ammocoete | 0 ($p = 1.0000$) | **157,630 ($p = 0.0001$)** | 15,638 ($p = 0.6586$) | 13,841 ($p = 0.0605$) | 8585 ($p = 0.0164$) |
| Control vs. Macropthalmia | 13,734 ($p = 0.47140$) | 43,255 ($p = 0.0897$) | 6425 ($p = 0.8502$) | 5936 ($p = 0.3608$) | 4801 ($p = 0.1179$) |

## 4. Discussion

Pacific lamprey provide multiple ecosystem benefits to freshwater systems across the species' life-stages. One benefit is the supply of marine-derived nutrients from adults that migrate upstream from the ocean to spawn and then die, providing nutrient loads for freshwater macroinvertebrates [31]. Adult lamprey also represent prey to larger predators (birds, bears, raccoon, etc.), and can potentially buffer predation on other species, e.g., salmonids [32]. The construction of redds by adults creates microhabitats for aquatic invertebrates [33], and aeration of the benthos by burrowing ammocoetes supports increased aerobic activity, which is associated with increased biodiversity of macroinvertebrates [34].

This study presents evidence of yet another, unappreciated ecosystem benefit generated by Pacific lamprey: potential water-quality enhancement due to the filter-feeding behavior of the ammocoete larval life-stage. The significant difference observed (Figure 2) in the rate of *E. coli* concentration declines between the 'Ammocoete' treatment and 'Control' treatment groups in the current experiment supports the idea that the presence of the filter-feeding ammocoete stage of Pacific lamprey can affect suspended concentrations of coliform bacteria. What is more, the finding that *E. coli* concentrations declined more rapidly in the presence of ammocoetes than when a similar density of lamprey macropthalmia was present suggests that the presence of the ammocoete life-stage, and not just lamprey in general, was crucial to the faster reduction in *E. coli* in ammocoete-containing aquaria. We hypothesized that the suspension-feeding behavior of the ammocoete life-stage would reduce suspended bacterial concentrations to improve water quality. Stemming from that proposition, the speed of reduction in *E. coli* might be expected to correspond with ammocoete number, as more feeding ammocoetes would be expected to more rapidly consume

suspended particulates. Supporting that prediction, the aquarium containing the higher density of ammocoetes showed the most rapid decline in *E. coli* concentrations, with *E. coli* abundance reaching the < 5% initial concentration threshold on day 3 of the experiment, 2 days earlier than any of the other experimental tanks. Although available fish numbers meant that only one ammocoete higher-density (4.3 fish·L$^{-1}$) tank was tested, that *E. coli* concentration declined most rapidly in that single aquarium suggests that environments with higher densities of ammocoetes can show more rapid clearing of *E. coli* from the water column. Taken as a whole, the findings presented here suggest that the presence of suspension-feeding ammocoetes can reduce the concentration of *E. coli* bacteria in the water, supporting the idea that the presence of lamprey may help promote the maintenance or recovery of water quality in a river system.

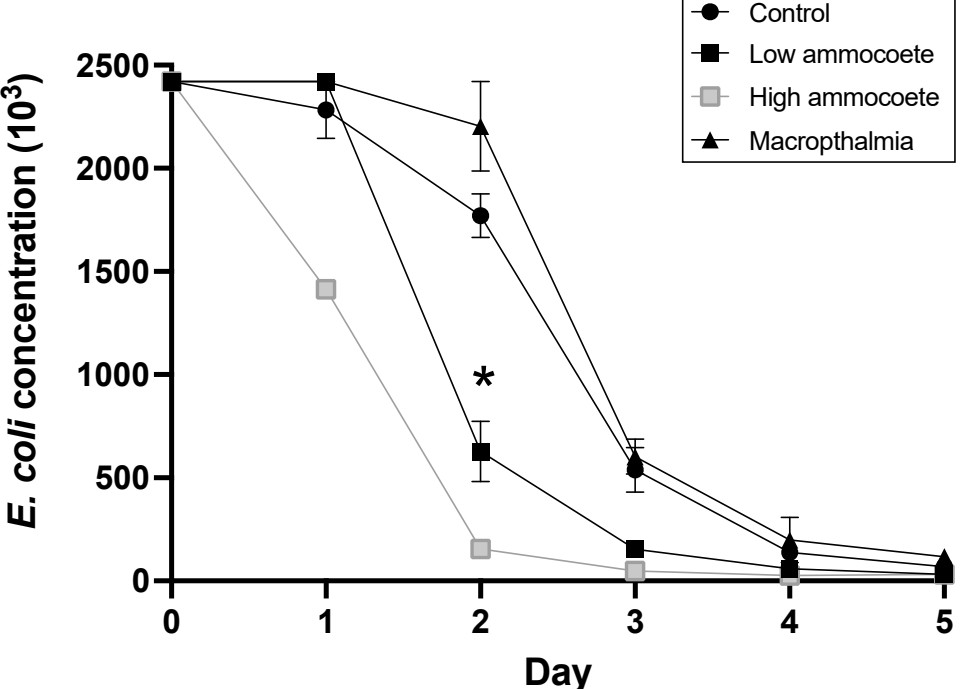

**Figure 2.** Changes in *E. coli* concentration values (mean ± standard error) over the 5 d study period. On day 2, the 'Ammocoete' treatment had a significantly lower viable *E. coli* concentration than either the 'Macropthalmia' or 'Control' treatments. The * indicates the day upon which significantly different estimates between 'Control' and 'Ammocoete' as well as between 'Macropthalmia' and 'Ammocoete' were observed. Note, the 'Ammocoete' treatment was not replicated and therefore not included in the statistical analyses. Note that standard error values for the 'Ammocoete low' treatment on day 3 lie within the point.

The morphological feeding mechanics of larval-stage lamprey species across the Order Petromyzontiformes is highly conserved [13]. Mallat [13] reported that the size of the particulate matter consumed by the sea lamprey (*Petromyzon marinus*) ranged from 4 to 200 μm in diameter. This size range was determined by pipetting the freshwater green alga *Chlorella pyrenoidosa* as well as pulverized ink in front of larval *P. marinus* and then dissecting the lamprey specimens, which revealed the ink particles and algae within the mucosal membranes of their pharynx and within their gut [13]. *E. coli* is a typical Gram-negative rod bacterium that has dimensions 1–2 μm long, with radius ~0.5 μm [35]. While this is smaller than the particulate sizes documented to be consumed by *P. marinus* [13], *E. coli* cells are commonly not just free-floating in water but also attached to suspended organic matter (e.g., detritus) of larger size, which would be consumed via filter-feeding by ammocoetes [36]. In another laboratory study it was shown that bacterial species, including Gram negative bacteria in the same family as *E. coli* (Family *Enterobacteriaceae*), within the

gut content of ammocoetes matched those in the water column but occurred at higher abundance [37]. Those findings combined with our observations collectively suggest that lamprey fishes broadly can remove small particulates from the water column, including suspended matter containing *E. coli*, when filter-feeding during their ammocoete phase.

Elevated levels of *E. coli* and other coliform bacteria in a watershed can result from contamination inputs that occur in periodic pulses or continuously [38]. For example, in San Luis Obispo Creek—the source for water used in the current experiment—rain events generate pulsed inputs of *E. coli* from urban run-off [23], while a localized pigeon population inhabiting a tunnel containing San Luis Obispo Creek likely contributes a more continuous input. Our current experiment simulated a pulsed input of *E. coli* because aquaria were filled at the start of the experiment with water containing a high initial concentration of *E. coli* and that initially bacteria-laden water was not renewed at any point during the 5 d experimental period. As expected given the experimental design, *E. coli* declined in all the treatment aquaria, but the 'Ammocoete' treatment demonstrated an faster decline in *E. coli* concentration. What remains untested, however, is whether ammocoetes can actually reduce *E. coli* concentrations in environments with a sustained input of *E. coli*. Sustained *E. coli* inputs are common in anthropogenically-impacted freshwater systems (e.g., streams coursing through urban areas or subject to agricultural runoff), and future research should address whether the presence of Pacific lamprey can reduce coliform bacterial levels in a larger context to support watershed quality management. Interestingly, suspension feeding by larval lamprey ammocoetes occurs with one of the slowest rates of water flow of any filter-feeding organism [13]. Feeding lamprey ammocoetes move water across the pharynx to trap microscopic particles at a water flow rate of ~30 mL·g$^{-1}$·h$^{-1}$, which is considerably slower than measured pumping rates of ~800 mL·g$^{-1}$·h$^{-1}$ for filter-feeding bivalve molluscs, ~700 mL·g$^{-1}$·h$^{-1}$ for sponges, or even the 225 mL·g$^{-1}$·h$^{-1}$ rate recorded for tunicates [9]. Given those slow rates of water pumping by ammocoetes, whether any effects of lamprey on water quality occur on a scale large enough to detectable empirically will depend on several factors including the density of ammocoetes present in the sediments of the aquatic system, the flow rate and water volume of the river or stream, and the extent and rate of sustained bacterial input into the system. All of those factors can vary considerably across time, as well as spatially in coastal stream or river systems, which are typically heterogenous in substrate and water flow. Field studies have shown that Pacific lamprey ammocoete density is patchy in coastal streams [39,40], and densities have been recorded to reach 120 fish·m$^{-2}$ in areas with preferred habitat features of fine substrate (mud, silt) and eddies or other areas of lower water flow [41]. Thus, while our data here provide evidence that the presence of lamprey ammocoetes can have effects on water bacterial concentrations, any such effects in a wild stream or river system are likely to be more complex and highly condition-dependent.

Indeed, the real test of Pacific lamprey ammocoetes as water quality-enhancing ecosystem service providers may ultimately require controlled field experiments or comparative empirical studies conducted at larger spatial scales in situ in watersheds. Even so, the findings presented here point to the possibility that the presence of larval Pacific lamprey may promote water-quality enhancement in freshwater systems, therein creating an ecosystem service that could be valuable both ecologically and socioeconomically, and thus may provide an additional justification for managing sustainable Pacific lamprey populations. It may thus be the case that restoring populations of Pacific lamprey and other lamprey species in the wild may not only aid the long-term persistence of these distinctive fish taxa, but also have the additional benefit of helping local water-resource managers improve water quality in impacted coastal watersheds.

## 5. Conclusions

In conclusion, Pacific lamprey ammocoetes in this study have shown an ecological function of filter-feeding and removing harmful bacteria. This has implications for watershed management and highlights the necessity of endemic species accessibility and

retention in watersheds. The speed and capacity of ammocoetes filter feeding potential would benefit from further studies.

**Author Contributions:** Conceptualization, P.K. and F.O.; methodology, P.K., F.O. and C.W.; software, J.S.; validation, P.K., C.W. and S.C.L.; formal analysis, J.S.; investigation, P.K.; resources, F.O.; data curation, P.K.; writing—original draft preparation, P.K.; writing—review and editing, C.W.; visualization, S.C.L.; supervision, F.O.; project administration, F.O.; funding acquisition, F.O. All authors have read and agreed to the published version of the manuscript.

**Funding:** Funding for this project was provided by the City of San Luis Obispo.

**Institutional Review Board Statement:** This study was conducted in adherence with the Guidelines for the Use of Fishes in Research published in 2014 by the joint committee of the American Fisheries Society, the American Institute of Fishery Research Biologists, and the American Society of Ichthyologists and Herpetologists. Fish were collected under Scientific Collecting Permit #10287 as permitted and in accordance to the approved methods of the California Department of Fish and Wildlife.

**Data Availability Statement:** Data is available via P.K. and will be shared with Google Drive.

**Acknowledgments:** Michael Gates provided laboratory assistance. Stewart Reid provided critical reviews that improved the quality of this manuscript. Tenera Environmental provided support and time to complete this study.

**Conflicts of Interest:** The authors declare no conflict of interest.

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
