# Peer review of "Filter-Feeding Pacific Lamprey (Entosphenus tridentatus) Ammocetes Can Reduce Suspended Concentrations of E. coli Bacteria"

_fishes, doi:10.3390/fishes8020101_

Round 1

Reviewer 1 Report

This is a well designed study with a clear objective.  It’s very clearly presented

It’s unfortunate that the statistical analysis resulted in only a significant effect for one day for one treatment (Day 2, low ammocoete).  While I understand that the high concentrations couldn’t be achieved in replicates due to the lack of availability of more fish, it’s a bummer.  I do think that the authors appropriate modulated their language “suggests” when writing about the single replicate high concentration samples, but the overall study would have been so much stronger with replicates for all studies. Hindsight is 20/20, I suppose.  In the end, I do not believe that this is a fatal flaw of the study.  The data demonstrates significant effect of the ammocoetes and suggests that this effect is dependent on fish concentration.  This is compelling and valuable. 

I admit that I did not really dig in to the statistical analysis, but it’s hard to understand why there’s not significant difference on day 3 between control and low ammocoete. I know that eyeballing statistics is dangerous and that there are not error bars shown for the low treatment at this time point, but the fairly small error of mean of the control relative to the measured day 3 low mean value would make we revisit this.  What does a simpler statistical test of just these two values indicate?

I would like that the authors include some more quantitative estimations related to the feasibility of to scale clearance and interpretation of these results.  This will be particularly valuable in the discussion … “whether any effects of lamprey on water quality occur on a scale large enough to detectable empirically will be dependent on several factors including the density of ammocoetes present in the sediments of the aquatic system, flow rate and water volume of the river or stream, and the extent and rate of sustained bacterial input into the system. “ Within the treatment tanks, the observed time scale of 1-2 days seems consistent with the rates included in the manuscript:  high concentration: 40 g (biomass) x 30 ml/gram/hr x 24 hrs/day = 28800 mL = 29 L. So, perhaps it is reasonable that this quantity of fish filter such a large amount of the total volume in one day. With similar analysis, knowing flow rates, water volumes, bacterial input and organism concentration, the authors could perform some estimate of the possible magnitude of effect in the environment.

Minor comments:

I just don’t understand “… remains unknown if and in what effect size micro ….” Particularly the “in what effect size” phrase.  Even if this is grammatically correct, please change to something that’s easier to read, particularly since it’s the first sentence of the abstract.

Demonstrating and accelerating decline in E. coli concentration … change “accelerating” to “greater rate of” or “higher” or “faster”.  Acceleration should refer only to a change in rate over time, not between treatments.

Why aren’t there error bars on all of the data points (aside from high ammocoete, which I understand was only one replicate)?

Reviewer 2 Report

This study provides novelty information related to the ability of Pacific Lamprey as a filter-feeding organism. However, a major revision in the introduction and methods are mandatory. Pay more attention to experimental treatment design and the amount of data needed for statistical analysis. Please have a look at the attached file for more suggestions.

Round 2

Reviewer 2 Report

The authors have significantly changed the manuscript and provided an essential suggestion for further similar research.